# High-Content Imaging-Based Assay for SARS-CoV-2-Neutralizing Antibodies

**DOI:** 10.3390/vaccines12030236

**Published:** 2024-02-24

**Authors:** Vinícius Pinto Costa Rocha, Bruna Aparecida Souza Machado, Helenita Costa Quadros, Antônio Márcio Santana Fernandes, Bianca Sampaio Dotto Fiuza, Cássio Santana Meira, Vitória Torres Barbosa da Silva, Afrânio Ferreira Evangelista, Larissa Moraes dos Santos Fonseca, Roberto José da Silva Badaró, Milena Botelho Pereira Soares

**Affiliations:** 1Institute of Health Technology, National Industrial Learning Service—Integrated Manufacturing and Technology Campus, SENAI CIMATEC, Salvador 41650-010, Bahia, Brazil; brunam@fieb.org.br (B.A.S.M.); antonio.fernandes@fbter.org.br (A.M.S.F.); bianca.fiuza@fbter.org.br (B.S.D.F.); cassio.meira@fieb.org.br (C.S.M.); vitorres2015@outlook.com (V.T.B.d.S.); afranio.evangelista@fieb.org.br (A.F.E.); larissa.fonseca@fieb.org.br (L.M.d.S.F.); badaro@fieb.org.br (R.J.d.S.B.); 2Laboratory of Tissue Engineering and Immunopharmacology, Oswaldo Cruz Foundation, Gonçalo Moniz Institute—Fiocruz, Salvador 40296-710, Bahia, Brazil; helenita.quadros@fiocruz.br

**Keywords:** SARS-CoV-2, antibodies, neutralization, high-content screening, variants

## Abstract

The COVID-19 pandemic and the consequent emergence of new SARS-CoV-2 variants of concern necessitates the determination of populational serum potency against the virus. Here, we standardized and validated an imaging-based method to quantify neutralizing antibodies against lentiviral particles expressing the spike glycoprotein (pseudovirus). This method was found to efficiently quantify viral titers based on ZsGreen-positive cells and detect changes in human serum neutralization capacity induced by vaccination with up to two doses of CoronaVac, Comirnaty, or Covishield vaccines. The imaging-based protocol was also used to quantify serum potency against pseudoviruses expressing spikes from Delta, Omicron BA.1.1.529, and BA.4/5. Our results revealed increases in serum potency after one and two doses of the vaccines evaluated and demonstrated that Delta and Omicron variants escape from antibody neutralization. The method presented herein represents a valuable tool for the screening of antibodies and small molecules capable of blocking viral entry and could be used to evaluate humoral immunity developed by different populations and for vaccine development.

## 1. Introduction

To date, 50 vaccines have been approved, authorized, licensed, or granted emergency-use authorization in at least one country [1]. Vaccination is the most effective way to avoid the development of severe acute respiratory syndrome, as it was estimated that 14.4 million deaths in 185 countries were prevented during the first year of vaccination [2]. Several SARS-CoV-2 variants have emerged since the declaration of the coronavirus disease (COVID-19) pandemic. The RNA-based viral genome facilitates mutation acquisition over time, and some mutations may make the virus more transmissible. More transmissible viral strains will overtake the previous one by infecting people, regardless of vaccination status, since intramuscular administration does not induce effective mucosal immunity capable of blocking viral spread through the upper airways [3,4,5,6]. 

SARS-CoV-2 has undergone great diversification, with more than 1500 PANGO lineages identified during the pandemic [7,8]. Those presenting increased transmissibility, virulence, and immune evasion are classified by the World Health Organization as variants of concern (VoC). Five VoCs have been reported, in addition to their subvariants, at the time of this study [9]. Since the emergence of the Omicron variant (B.1.1.529), which harbors an unprecedented number of mutations, the monitoring of neutralizing antibodies has become even more necessary to determine a given population’s neutralizing capacity, mainly four to six months after vaccine administration, when antibody levels tend to decrease.

Viral particles carrying a reporter gene sequence and expressing the spike are important tools in terms of evaluating the neutralizing capacity of antibodies [10,11,12]. Muruato and colleagues (2020), for example, used a reverse genetic system to produce replicating recombinant SARS-CoV-2 expressing the mNeonGreen reporter gene [10,13]. When tested with human convalescent plasma, their assay was comparable to the gold standard plaque reduction neutralization test (PRNT) and is being used in a phase I/II clinical trial [10,14]. Other groups developed similar assays using the lentiviral approach, which has the advantage of being manipulated at BSL-2 laboratories [15,16,17,18,19,20,21]. Based on the importance of evaluating the neutralizing capacity of human serum and developing new vaccines, when considering the variation among different populations, we standardized and validated an imaging-based high-content assay, which has been shown to be safe, high-throughput compatible, and able to determine the neutralizing potency of plasma in two days, which is half of the time required by the PRNT [22]. Our assay is based on commercially available plasmids used to produce lentiviral non-replicating particles expressing spike glycoproteins, which can transduce ACE2-overexpressing human embryonic kidney cells (HEK293T). Infection results in the integration of the ZsGreen sequence into the cell genome. Automated image analysis employs a predefined algorithm that can detect fluorescent cells. This assay was shown to detect neutralizing antibodies in previously analyzed human samples regarding the production of total IgG and neutralizing antibodies following the administration of the first and second doses of three clinically used vaccines. Screening can be expanded to detect monoclonal antibodies and small molecules that are able to block viral infection and could also be used for vaccine development. 

## 2. Materials and Methods

### 2.1. Plasmids and Reagents

The plasmid encoding Spike Wuhan-Hu-1 was obtained from BEI Resources, as well as all packaging vectors for lentiviral production (NIAID, NIH: SARS-Related Coronavirus 2, Wuhan-Hu-1 Spike-Pseudotyped Lentiviral Kit V2, NR-53816). Delta B.1.617.2 (plasmid #179905), Omicron B.1.1.529 (plasmid #179907), and Omicron BA.4/5 (plasmid #186031) plasmids were purchased from Addgene (Watertown, MA, USA) [23,24]. All other general laboratory chemicals and solvents used were of analytical or HPLC grade. The plasmids pHAGE-CMV-Luc2-IRES-ZsGreen-W, HDM-Hgpm2, HDM-Tat1b, and pRC-CMV-Rev1b were obtained from BEI Resources (NIAID, NIH: SARS-Related Coronavirus 2, Wuhan-Hu-1 Spike-Pseudotyped Lentiviral Kit V2, NR-53816).

### 2.2. Ethics Statement

Thirty volunteers kindly participated in a cohort study aimed at analyzing humoral response prior to and after receiving two doses of SARS-CoV-2 vaccines (CoronaVac, Comirnaty, or Covishield) at the beginning of the vaccination campaign in Brazil between February and August 2021. The study was conducted in accordance with the Declaration of Helsinki and approved by the Ethics Committee of National Industrial Learning Service–Integrated Manufacturing and Technology Campus (SENAI-CIMATEC) (protocol code 4.334.505; date of approval: 2 September 2021). The samples were collected by convenience, following the provision of informed consent, from employees of SENAI-CIMATEC. The studied population consisted of men and women aged 18 years or older. All volunteers received results from serological testing performed using an Enzyme-Linked Immunosorbent Assay (ELISA), and antibody neutralization was conducted using an ELISA Kit from Elabscience (Houston, TX, USA).

### 2.3. Determination of Anti-SARS-CoV-2 IgG Antibodies

The ELISA for total IgG against the Spike protein followed the previously published protocol [25]. Serum samples from all participants were collected before vaccination, 30 days after receiving the first dose, and 30 days after the second dose. Briefly, 96-well flat-bottom microplates were sensitized with 50 µL of 1 µg/mL *spike* D614G recombinant protein (kindly provided by Sean Grey from PAI Life Sciences, Seattle, WA, USA) diluted in phosphate-buffered saline (PBS, pH 7.2), and subsequently incubated at 4 °C overnight. After 16 h of incubation, the microplates were washed three times with a washing solution containing PBS plus 0.05% Tween 20 (Sigma Aldrich, San Luis, MO, USA) and then blocked with a PBS solution containing 3% skim milk for 1 h. Fifty microliters of serum samples at an initial dilution of 1:10 was dispensed into the wells, followed by a two-fold serial dilution in PBS with 1% skim milk, ranging from 1:10 to 1:1280.

The detection of neutralizing antibodies was performed using the SARS-CoV-2 Neutralization Antibody ELISA Kit from Elabscience (Houston, TX, USA), following the manufacturer’s instructions. This kit, based on the competition ELISA technique, aims to detect the presence of neutralizing antibodies against SARS-CoV-2. All testing was performed using serum dilutions ranging from 1:40 to 1:2560.

### 2.4. Cell Cultures

Human Embryonic Kidney (HEK) 293T cell line and HEK 293T overexpressing ACE2 (293T-ACE2) were cultivated in Dulbecco’s Modified Eagle Medium (DMEM, Thermo Fisher, Waltham, MA, USA) supplemented with 10% FBS (Vitrocell, Campinas, São Paulo, Brazil), 50 U/mL of penicillin, 50 µg/mL of streptomycin (PenStrep; Thermo Fisher), and 100 mM of HEPES (Thermo Fisher). The cells were incubated at 37 °C and 5% CO_2_ in T25 or T75 vented cap flasks, with cell maintenance performed every 3 to 4 days by cell trypsinization (LGC Biotecnologia, Cotia, São Paulo, Brazil). The 293T ACE2 cells were obtained through BEI Resources (NIAID, NIH: Human Embryonic Kidney Cells Expressing Human Angiotensin-Converting Enzyme 2, HEK-293T-hACE2 Cell Line, NR-52511).

### 2.5. Generation of Pseudotyped Lentiviral Particles

Pseudotyped lentiviral particles expressing the *spike* Wuhan-Hu-1 protein (GenBank: NC_045512), Delta B.1.617.2, Omicron B.1.1.529, and BA.4/5 were generated via the co-transfection of the HEK 293T cell line, according to a previously described protocol [26]. For this, HEK 293T cells were plated on 6-well plates at 5 × 10^5^ in 2 mL of supplemented DMEM and then incubated at 37 °C under 5% CO_2_ for 24 h. Next, the cells were transfected using Lipofectamine 3000 (Thermo Fisher) and 2 µg of a DNA mix containing the following lentiviral constructs to produce the pseudotyped virus expressing the Wuhan-Hu-1 spike surface glycoprotein: a total of 1 µg of pHAGE-CMV-Luc2-IRES-ZsGreen-W, 0.22 µg of HDM-Hgpm2, 0.22 µg of HDM-Tat1b, 0.22 µg of pRC-CMV-Rev1b, and 0.34 µg of HDM-IDTSpikeΔCter. For pHAGE-CMV-Luc2-IRES-ZsGreen-W, a map and the sequence are provided in Appendix A, respectively. To produce Delta, Omicron B.1.1.529, or BA.4/5 pseudotyped particles, the plasmid encoding Wuhan-Hu-1 spike glycoprotein was replaced by 0.34 µg of pTwist-SARS-CoV-2 Δ18 B.1.617.2v1, pTwist-SARS-CoV-2 Δ18 B.1.1.529, or pCAGGS SARS-CoV-2 BA.4/5 Spike, respectively [23,24]. After 18 h of transfection, the media was replaced in each well, and the plates were incubated for an additional 48 h. Sixty hours after transfection, the supernatants containing the pseudoviruses were collected and filtered using a 0.45 µm PES membrane filter. Pseudovirus preparations were titrated and used in neutralization assays. Viral particles with no entry protein (“bald” pseudoviruses) were used as negative controls in the titration assays, while a pseudotyped particle expressing the vesicular stomatitis virus G (VSV G) protein was used as a positive control.

### 2.6. Assay Precision

Repeatability (intra-assay) was determined by one researcher through the performance of three independent experiments (three viral batches) in triplicate. Reproducibility (inter-assay) was determined by two researchers who performed three independent experiments with five replicates [27]. The coefficient of variation (CV) was calculated by the ratio between the standard deviation and the pNT_50_ mean values. The CV up to 15% was accepted for both precision parameters [28]. 

### 2.7. Cell Transduction, Image-Based Analysis, and Titration

Using a 96-well flat-bottom microplate, the produced pseudovirus was 2-fold serially diluted in supplemented DMEM at a final volume of 100 µL per dilution, which resulted in seven concentrations, including the non-diluted stock solution. The 293T-ACE2 cells were added to a 96-well plate at a density of 1.25 × 10^4^ in 50 µL of supplemented DMEM and then incubated at 37 °C under 5% CO_2_ for 48 h. Cell nuclei were labeled using Hoechst 33342 (Abcam, Cambridge, UK) at 20 µM and, after 15 min of incubation at 37 °C, the percentage of 293T-ACE2 cells expressing ZsGreen was quantified on a CellInsight CX7 LZR platform (Thermo Fisher) and analyzed using Cellomics Scan software (version 6.6.2).

The algorithm used to quantify ZsGreen-positive cells was based on the Cellomics Scan Target Activation protocol, resulting in 25 analyzed fields per well (the entire well). To identify the primary objects using the Hoechst 33342 channel, the algorithm applied a mask in the nuclear region. Primary object validation and selection were performed in accordance with size and fluorescence intensity. Objects validated using the Hoechst 33342 channel were then used to determine the number of identified cells per well. To identify objects using the ZsGreen channel, a mask was applied to the cytoplasm region, excluding the nucleus. Object validation and selection were performed in accordance with fluorescence intensity by performing comparisons among 293T-ACE2 plus pseudovirus groups, 293T-ACE2 plus the “bald” pseudovirus, and 293T-ACE2 cells only. Reference levels were established based on negative controls, with average fluorescence intensity over the reference used to determine the percentage of ZsGreen-positive cells. Pseudovirus titers were determined using the Poisson formula, which revealed the number of viral particles per milliliter in accordance with the following equation: −ln (1 − P/100) × (number of cells per well)/(volume of pseudovirus added per well in milliliters), where P is the percentage of ZsGreen^+^ cells [26].

### 2.8. Neutralization Assay

To validate this new method, the same samples from 30 subjects were used after being collected at three different times: before vaccination, 30 days after the first immunization, and 30 days after the second immunization. Using a 96-well microplate, 5 µL of serum from each subject was added to wells A1 through A12, followed by the addition of 95 µL of supplemented DMEM medium, corresponding to a 1:20 serum dilution. From well B1 to G12, 50 µL of supplemented DMEM was added in the absence of serum. The samples were serially diluted by taking 50 µL from the sample diluted at 1:20 and passing it to the next well below in the same column. Twelve wells (H1 to H12) were left without serum, and 50 µL of supplemented DMEM was added. Fifty microliters of the pseudotyped particles was added to each well already containing 50 µL of diluted serum, corresponding to a final dilution curve ranging from 1:40 to 1:2560. The wells corresponding to the negative and positive controls for neutralization received 50 µL of spike-expressing pseudoviruses (H1 to H6) or control bald pseudoviruses (H7 to H12), respectively. The plates were then incubated at 37 °C under 5% CO_2_ for 1 h to allow the antibodies to interact with the spike. Next, 293T-ACE2 cells were added at a density of 1.25 × 10^4^ in 50 µL of supplemented DMEM, followed by an additional 48 h of incubation at 37 °C under 5% CO_2_. Hoechst 33342 staining and image analysis were performed as described above. The serum titration curve was used to determine the half-maximum effective concentration of each sample against the pseudoviruses (pseudovirus neutralization—pNT_50_). This parameter corresponds to the serum dilution capable of neutralizing 50% of the pseudoviruses and is represented by the reciprocal value (1/pNT_50_), i.e., higher values indicate greater sample potency.

### 2.9. Statistical Analysis

Statistical analyses were performed using GraphPad Prism 8.0 (Boston, MA, USA). Data normality was evaluated by using the Shapiro–Wilk test. Statistical differences among the three groups were evaluated using one-way ANOVA or Friedman test, with results considered significant when *p* < 0.05. Statistical differences between the two groups were evaluated via the Mann–Whitney test and considered significant when *p* < 0.05. pNT_50_ was determined via nonlinear regression by the Log_10_ from serum dilutions and the percentage of reduction in infection.

## 3. Results

### 3.1. Standardization of a High-Content Imaging-Based Neutralization Assay

To standardize the high-content imaging-based neutralization assay using 293T-ACE2 cells, we first produced lentiviral particles expressing the spike glycoprotein from the Wuhan-Hu-1 variant (GenBank: NC_045512). Viable viral particles transduced the target cells and integrated the ZsGreen sequence in the host genome, generating a green fluorescent transgenic cell line. Using the Target Activation protocol in Cellomics software, the percentage of ZsGreen-positive cells (ZsGreen^+^) was automatically quantified (Figure 1a). No fluorescent cells were identified via transduction with control pseudoviruses not expressing the spike protein (bald virus, Figure 1b), as only cells transduced by spike-expressing pseudoviruses exhibited fluorescence (Figure 1c). Cell nuclei were visualized using the Hoechst 33324 channel, while ZsGreen^+^ cells were identified by using the algorithm’s region of interest in the cytoplasm. Average ZsGreen intensity was used to separate populations exhibiting negativity or positivity for ZsGreen (Figure 1d–g). Using a viral dilution curve, we identified significantly more ZsGreen^+^ objects in the groups infected with undiluted viral spike (S) stock or 1:10 VSV-G-expressing pseudoviruses, as well as among the dilutions, characterizing a titer detection relationship. HEK293T cells were only able to be infected by VSV-G-expressing pseudovirus, confirming the need for ACE2 in the model involving spike-expressing particles (Figure 2a,b). Linear regression curves are shown in Appendix A. The reduction in ZsGreen^+^ cells in relation to viral titers is evidenced in Figure 2c–g. Moreover, we established correlations between not only the percentage of ZsGreen^+^ cells and the number of pseudotyped viral particles per milliliter but also between average ZsGreen fluorescence intensity and the number of viral particles/mL (Figure 2h,i). We observed a linear trend between both parameters and viral titers. A Pearson’s correlation coefficient (r) value of 0.9961 was found in relation to the percentage of ZsGreen^+^ cells and 0.9865 regarding ZsGreen average fluorescence intensity (Appendix A). As the mean ZsGreen intensity was used to select ZsGreen^+^ cells, the cut-off was calculated as the mean of ZsGreen intensity from negative controls plus 3× the standard deviation. The cut-off value was calculated at 39 (a.u.); however, it is important to highlight that we determined the cut-off for each analyzed plate based on the positive control for neutralization (the lowest ZsGreen percentage) represented by infection with pseudoviruses without entry proteins. The assay precision was determined by intra- and inter-assay variability analysis using two WHO standard samples. Fifty microliters of fresh pseudoviruses were used, corresponding to a multiplicity of infection (MOI) of approximately 0.6. The CV measured in the intra-assay research was 14% and 11.9% for 21/388 and 21/340, respectively. Two researchers independently tested the WHO standards to determine the inter-assay reproducibility. The CV calculated between the researchers was 19% for 21/338 and 3.2% for 21/340, demonstrating less than 15% of variability (Figure 3a,b), except for the inter-assay regarding sample 21/388. Nonlinear regression curves are shown in Figure 3c,d.

### 3.2. Imaging-Based Analysis to Detect the Neutralizing Capacity of Volunteers after Two Vaccination Doses

To validate the imaging-based protocol, serum samples from cohorts of volunteers vaccinated with two doses of three different vaccines were used. Of the 30 participants, 15 (50%) were immunized with the Covishield vaccine, representing the largest study group, followed by 8 (26.7%) participants vaccinated with Comirnaty, and 7 (23.3%) with CoronaVac (Figure 4a). Regarding participant sex, females predominated (*n* = 22, 75.9%). The predominant race in the three subgroups was white (*n* = 14, 48.3%), followed by black (*n* = 7, 24.1%). Only two (6.9%) individuals tested positive for COVID-19 via RT-PCR prior to vaccination. At the time of vaccination, these volunteers were already negative for COVID-19, as determined by RT-PCR. Firstly, the samples were evaluated using a neutralization assay kit using recombinant HRP-conjugated receptor-binding domain (RBD) of the Wuhan-Hu-1 spike glycoprotein. The percentage of neutralizing antibodies was determined in all samples at a 1:40 dilution. As shown in Figure 4, immunization with CoronaVac induced 19.83% and 60.63% neutralization after the first and second doses, respectively. Immunization with Comirnaty and Covishield induced 20.36% and 34.56% neutralization after the first dose, with 84.61% and 71.38% following the second dose, respectively (Figure 4b–d). Serum potency (presented as pNT_50_) was determined via nonlinear regression. Samples from volunteers vaccinated once with CoronaVac presented pNT_50_ values at the detection limit of 40, as those prior to immunization and after the first dose. Only after receiving the booster did pNT_50_ increase to 53.61. Samples from volunteers vaccinated with Comirnaty presented a slight increase (pNT_50_ = 44.96) after the first dose and an approximately six-fold increase after the second dose (pNT_50_ = 224.88). Volunteers vaccinated with Covishield presented respective pNT_50_ values of 40, 69.46, and 151.40 before immunization, after the first, and following the second dose (Figure 4e–g). In addition, total IgG titers demonstrated an increase in antibody production after the second dose (Appendix A).

These samples were submitted to the imaging-based protocol to determine the potency of neutralizing antibodies. Pseudoviruses at MOI 0.6 were used for each spike variant, resulting in 34.4 ± 7.2% of ZsGreen^+^ cells for wild-type virus, 26.2 ± 13.8% for Delta, 43.6 ± 6.5% for Omicron B.1.1.529, and 9.8 ± 1% for Omicron BA4/5. This viral amount was enough to produce at least a 30-fold increase in the ZsGeen mean fluorescence intensity, which can distinguish transduced and non-transduced cells. The samples were tested at a 1:40 dilution to determine the percentage of neutralization against the Wuhan-Hu-1 spike glycoprotein pseudovirus. Samples from volunteers prior to vaccination presented a maximum level of 32% pseudovirus neutralization. The first dose of Covishield significantly increased neutralization. A similar profile was not observed after receiving the first dose of CoronaVac and Comirnaty. Following the second dose, Comirnaty and Covishield significantly increased neutralization capacity compared to before vaccination (Figure 5a–c). Serum potency was determined via nonlinear regression and is represented as pNT_50_. Samples from volunteers vaccinated with CoronaVac presented pNT_50_ at the detection limit of 40, with enhanced serum potency of 213 only following the second dose. However, based on paired analysis, the second dose of CoronaVac did not significantly increase the neutralization capacity of the sera. Samples from naïve volunteers before receiving Comirnaty presented a potency at the detection limit, which increased to 265 after the first dose. In these individuals, pNT_50_ was significantly higher than before vaccination after receiving a second dose of Comirnaty. A similar profile was found in samples of volunteers vaccinated with Covishield. In this case, pNT_50_ significantly increased from 40 to 231 after the first dose and remained at 203 after the second dose (Figure 6a–c). Nonlinear regression curves are shown in Appendix A. These data are consistent with the neutralizing antibody measurements obtained using ELISA despite slight increases in pNT_50_ mean values detected by the imaging-based assay (Appendix A).

The imaging-based protocol was also used to analyze the neutralizing ability of the samples against the main VoCs (at the time of this research), including Delta (B.1.617.2), and Omicron B.1.1.529 and B.A4/5. Our results show that vaccination with CoronaVac did not induce antibodies capable of neutralizing these VoCs (Figure 6d,g,j). The mRNA and vector-based vaccines were found to significantly increase the levels of serum-neutralizing potency against the Delta VoC only following the second dose (Figure 6e,f). These vaccines did not significantly increase serum potency against Omicron B.1.1.529 (Figure 6h,i) and BA.4/5 (Figure 6k,l). Nonlinear regression curves are shown in Appendix A.

We observed that the ability of the vaccines to induce neutralizing antibodies against VoCs became reduced following the evolution of the variants. Following the second dose of CoronaVac, the calculated pNT_50_ showed a three-fold and a five-fold lower value, respectively, when the serum samples were subjected to neutralization assays using Delta and omicron VoC spikes, relative to the pNT_50_ calculated using pseudoviruses expressing Wuhan-Hu-1 spike (Figure 7a). The same profile was observed regarding the wane of serum potency following one and two doses of Comirnaty or Covishield (Figure 7b,c). These data demonstrated the antibody neutralization escape of VoCs detected using our imaging-based protocol. It is noted that some volunteers presented high-potency serum against Delta after the second dose of Comirnaty and Omicron B.1.1.529 after the first and second doses of Covishield.

## 4. Discussion

In this work, we standardized a method of automated imaging analysis to perform a neutralization assay using pseudovirus. The pseudotyped viral particles produced were functional and capable of transducing ACE2-overexpressing cells. The protocol for automated imaging analysis can detect different levels of viral titers during the infection assay presenting accepted levels of variability among the experiments. Moreover, the protocol can also detect changes in serum titers induced by vaccination and measure the potency of neutralizing antibodies. Therefore, this protocol represents a useful tool for screening neutralizing antibodies and small molecules to prevent SARS-CoV-2 VoC infection, as well as for vaccine development [29,30]. Our method presented similar results on samples previously characterized by using ELISA with respect to the neutralization of Wuhan-Hu-1 RBD. Moreover, our results were near to those reported in previous studies [23,31]. It is noteworthy that slight differences in the pNT_50_ neutralization results were measured by using ELISA and our imaging-based method. It can be explained by the higher sensitivity of the imaging-based assay with respect to neutralizing antibodies targeting the S1 spike domain outside the RBD [32]. As the commercial kit employs RBD, no neutralizing antibodies outside this region are not detected.

Different groups standardized methods like ours, which were also demonstrated to produce results in agreement with the PRNT and identify neutralizing escape when the samples were tested against VoCs [15,16,17,18,19,20,21,23,31]. PRNT requires a BSL-3, rendering it non-safety compatible since it requires live viruses, is laborious, time-consuming, and is not well adapted to high-throughput studies [22,33]. Reverse genetics was used to develop SARS-CoV-2 particles carrying a reporter gene; however, it still needs BLS-3 infrastructure to handle this transgenic virus and is incompatible with most laboratories around the world [10]. It reflects the importance of this type of neutralization assay presented here. The use of pseudovirus presents the disadvantage of being restricted to the study of viral entry mechanisms since it is non-replicating. However, pseudoviruses are generally stable compared to wild-type viruses, which need to be propagated in animal cells. Each interaction with the host cells may generate point mutations, meaning that the viral batch tested is not homogeneous. With the use of plasmids, it is more controlled, and viral particles contain a unique clone of spike glycoprotein. Furthermore, the use of plasmids eases the assays with viral variants. As demonstrated in this work, vectors for expressing mutated spikes are interchanged to produce new pseudoviruses.

The continuous emergence of VoCs highlights the importance of regularly monitoring the population’s neutralizing capacity, as well as evaluating vaccine efficacy against newly evolving viral strains. BQ.1.1 and XBB.1, for example, have demonstrated high resistance against vaccine-induced neutralizing antibodies. The neutralizing antibody titer against BQ.1.1 and XBB.1 was shown to decrease by 53- and 127-fold, respectively, compared to the Wuhan-Hu-1 strain after monovalent boost (two doses followed by monovalent booster) and by 80- and 232-fold after bivalent boost (two doses followed by bivalent booster), respectively [34]. 

In this work, we used the imaging analysis to evaluate pseudovirus neutralization; however, a pseudovirus based on pHAGE-CMV-Luc2-IRES-ZsGreen-W vector also integrates luciferase reporter gene in the host cell, making it possible to validate neutralization by using other methods, such as a bioluminescence plate reader [26,27]. Therefore, neutralizing protocols like ours should be widespread in different countries to monitor and characterize the immune response by neutralizing antibodies in a specific population. 

It is noteworthy that even though the number of subjects evaluated herein was small, it was not our intention to describe biological patterns in the volunteers. Rather, we intended to use the samples to evaluate if our imaging-based method could provide consistent results through the quantification of serum potency. We also showed that some volunteers could efficiently neutralize Delta or Omicron B.1.1.529. The reason for this high neutralizing capacity was not investigated and is unknown. This can be a consequence of VoC infection, or they have naturally high responsiveness regarding the production of neutralizing antibodies against Delta or Omicron B.1.1.529. For the evaluation of the results produced by using the imaging-based protocol, the cohorts used were of sufficient size to generate data that were close to those reported in previous publications [23]. 

## 5. Conclusions

In sum, the method presented in this work was shown to efficiently detect changes in neutralizing antibody titers induced by vaccination. This pseudovirus imaging-based assay represents a valuable tool in the fight against COVID-19, which is presently focused on the vigilance of vaccine-induced neutralizing antibodies against newly emerging variants and subvariants, the development of new COVID-19 vaccines, as well as the screening of compounds and antibodies designed to prevent or reduce viral infection. Furthermore, this method can be used to monitor the waning of protective neutralizing antibodies acquired by infection or vaccination. 

## Figures and Tables

**Figure 1 vaccines-12-00236-f001:**
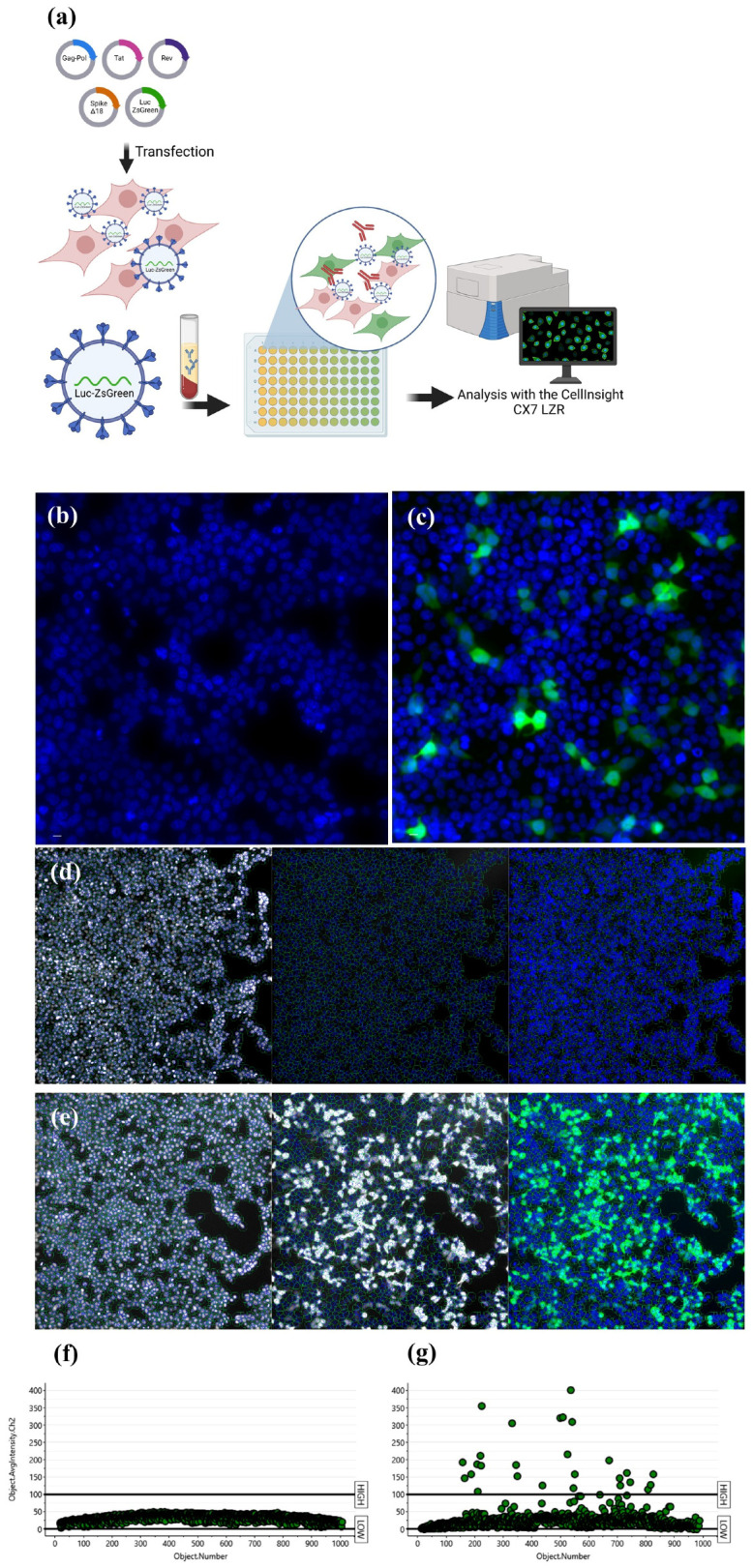
Automated imaging-based analysis workflow. (**a**) HEK-293T cells were transfected with plasmids to generate pseudoviruses. Viral particles were collected from supernatants and used in neutralization assays using HEK-293T-ACE2 cells together with serum from volunteers. The quantification of ZsGreen^+^ cells was determined automatically via Cellomics software using a CX7LZR high-content system. (**b**) Pseudoviruses without an entry spike protein (bald virus) did not generate fluorescent cells, (**c**) while spike expression promotes viral entry into ACE2-overexpressing cells. (**d**,**e**). The Target Activation protocol from Cellomics was applied to detect cell nuclei and determine the region of interest (ROI) to detect the ZsGreen fluorescence. Representative imaging analysis of the average fluorescence intensity used to separate the cellular population of (**f**) ZsGreen^−^ and (**g**) ZsGreen^+^ cells. The cut-off established as 100 was only for representative purposes. The cut-off is finely adjusted in each assay. Nuclei was stained in blue with Hoechst and ZsGreen fluorescence is shown in green.

**Figure 2 vaccines-12-00236-f002:**
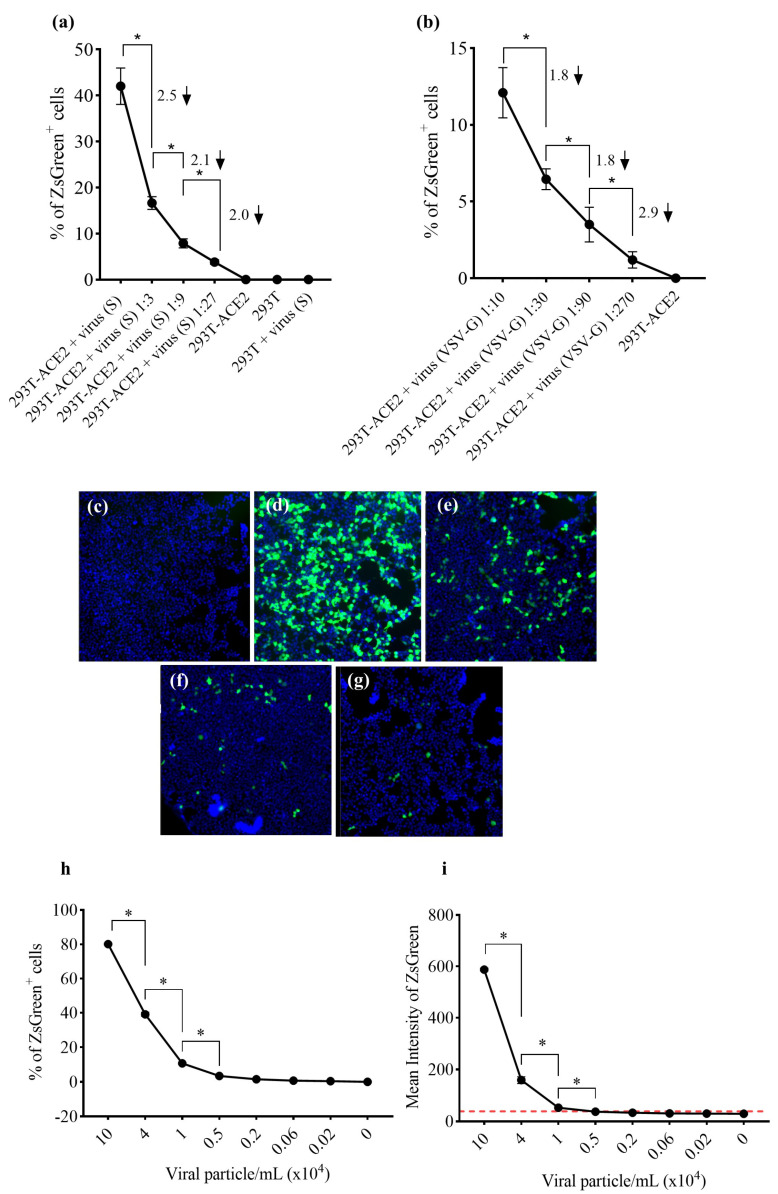
Detection of viral titers via automated imaging analysis protocol: the percentage of ZsGreen^+^ cells and average fluorescence intensity were measured by using the image-based analysis protocol. (**a**) Pseudoviruses expressing the Wuhan-Hu-1 spike glycoprotein are functional, as only HEK293T-ACE2 cells were transduced. Numbers above the graphic line and the arrows denote the reduction fold calculated by the ratio between viral amounts. (**b**) VSV g-expressing particles were used as a transduction control since both cell lines express the receptor for this ligand. Numbers above the graphic line denote the reduction fold calculated via the ratio between viral amounts. Fluorescence images of (**c**) bald pseudoviruses, (**d**) undiluted Wuhan-Hu-1 spike-expressing pseudoviruses, (**e**) Wuhan-Hu-1 pseudoviruses diluted 3-fold, (**f**) diluted 9-fold, and (**g**) diluted 27-fold, as illustrated by decreasing numbers of ZsGreen^+^ cells. The protocol was shown to both (**h**) detect decreasing levels of Wuhan-Hu-1 pseudovirus titers based on the percentage of ZsGreen^+^ and (**i**) on the average fluorescence intensity. The red line represents the calculated cut-off value (39 a.u.). Data are shown as mean ± SD. * One-way ANOVA, followed by Holm–Sidak’s multiple comparisons test, *p* < 0.05, calculated by using GraphPad Prism, version 8.0.

**Figure 3 vaccines-12-00236-f003:**
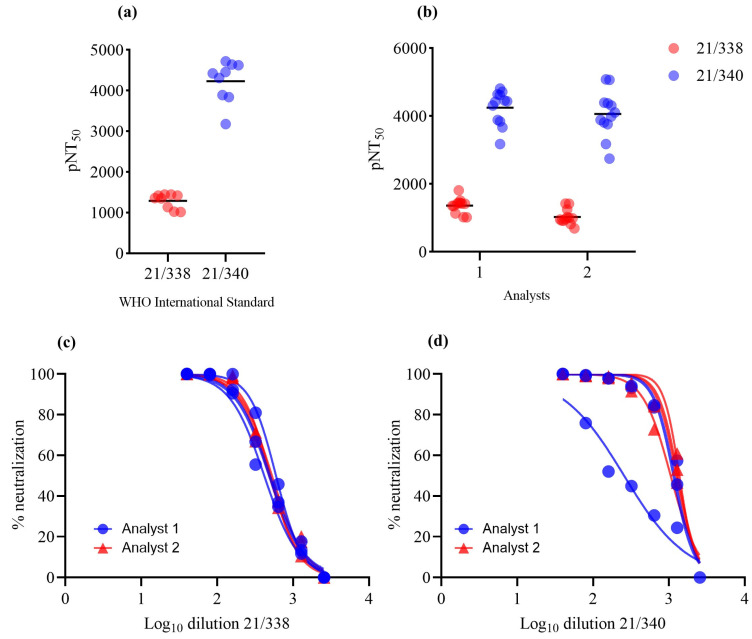
Precision tests of the automated imaging analysis. Pseudoviruses expressing the Wuhan-Hu-1 spike glycoprotein were included with WHO 21/338 and 21/340 standard serum samples. The pNT50 was calculated, as well as the CV in the same (intra-assay) or independent experiments (inter-assay). (**a**) Intra-assay: 3 independent experiments were performed by one researcher. Data were pooled to represent the variation among the samples. (**b**) Inter-assay analysis: 2 researchers performed 3 independent assays. Data were pooled to represent the variation among the samples. (**c**,**d**) Nonlinear curves regarding the precision assays.

**Figure 4 vaccines-12-00236-f004:**
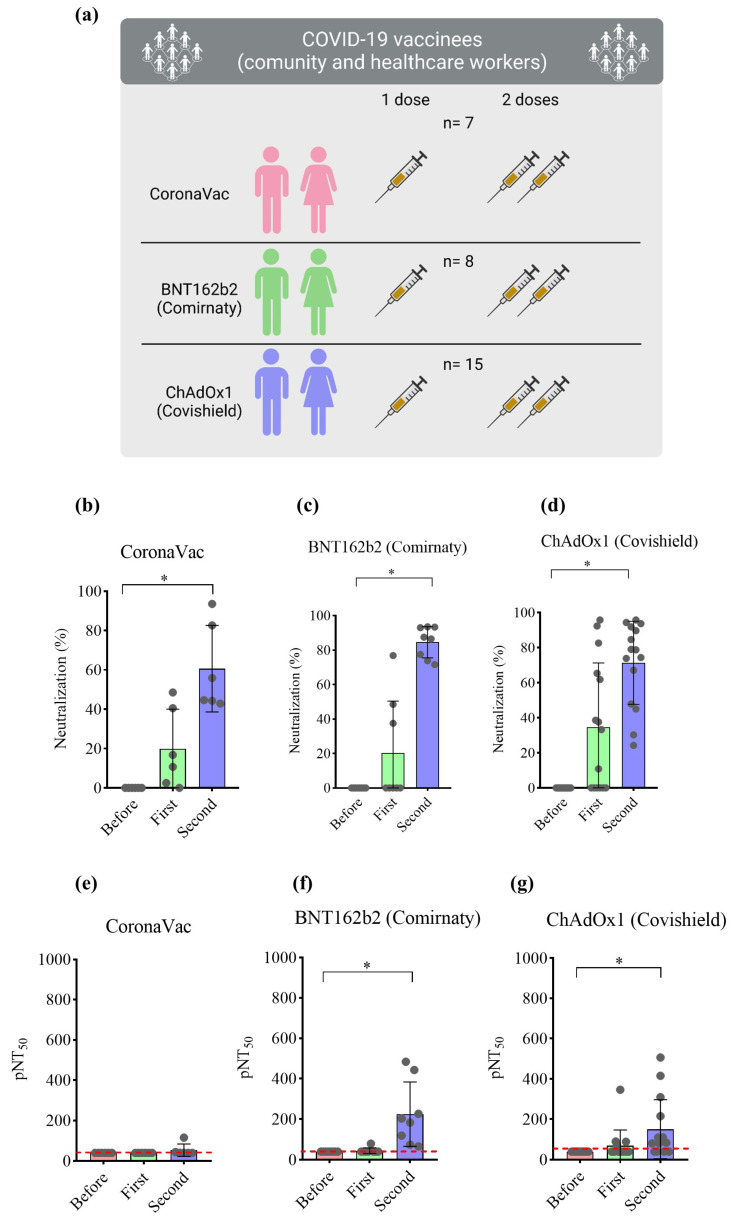
Percentage of neutralization and pNT_50_ values in volunteers immunized with different vaccines. (**a**) Schematic representation of the cohorts of volunteers tested. Blood from healthy volunteers was collected prior to and 30 days after the first and second doses of three different vaccines against SARS-CoV-2. Serum was collected and stored at −80 °C until use. Samples were tested with neutralization assay kit using recombinant HRP-conjugated RBD of the Wuhan-Hu-1 spike glycoprotein. (**b**–**d**) Percentage of neutralization of serum from volunteers immunized with CoronaVac (*n* = 7), Comirnaty (*n* = 8), or Covishield (*n* = 15), respectively, at 1:40 dilution. (**e**–**g**) pNT_50_ of volunteers immunized with CoronaVac (*n* = 7), Comirnaty (*n* = 8), or Covishield (*n* = 15), respectively. Values considered statistically significant when *p* < 0.05 (*) via the Friedman test, followed by Dunn’s multiple comparisons test, calculated by using GraphPad Prism, version 8.0. Data are shown as mean ± SD. Circles indicate the volunteers, and red line indicate the lowest pNT_50_ value equal to 40.

**Figure 5 vaccines-12-00236-f005:**
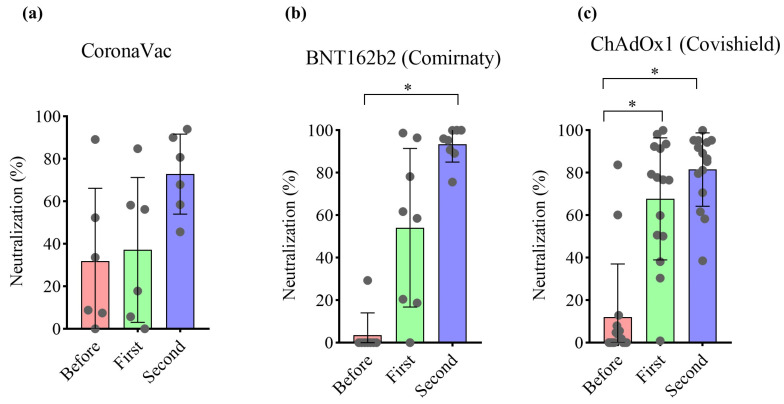
Imaging-based analysis detects the neutralizing capacity of volunteers after vaccination and following boost with (**a**) CoronaVac, (**b**) Comirnaty, or (**c**) Covishield against Wuhan-Hu-1 spike glycoprotein pseudovirus. Samples were submitted to the neutralization image-based assay for method validation. The neutralization capacity of serum specimens was determined by the transduction of HEK293T-ACE2 cells using Wuhan-Hu-1 spike-expressing pseudovirus. Values were considered statistically significant when *p* < 0.05 (*) examined using Friedman test, followed by Dunn’s multiple comparisons test, calculated by GraphPad Prism, version 8.0. Data are shown as mean ± SD. Circles indicate the volunteers.

**Figure 6 vaccines-12-00236-f006:**
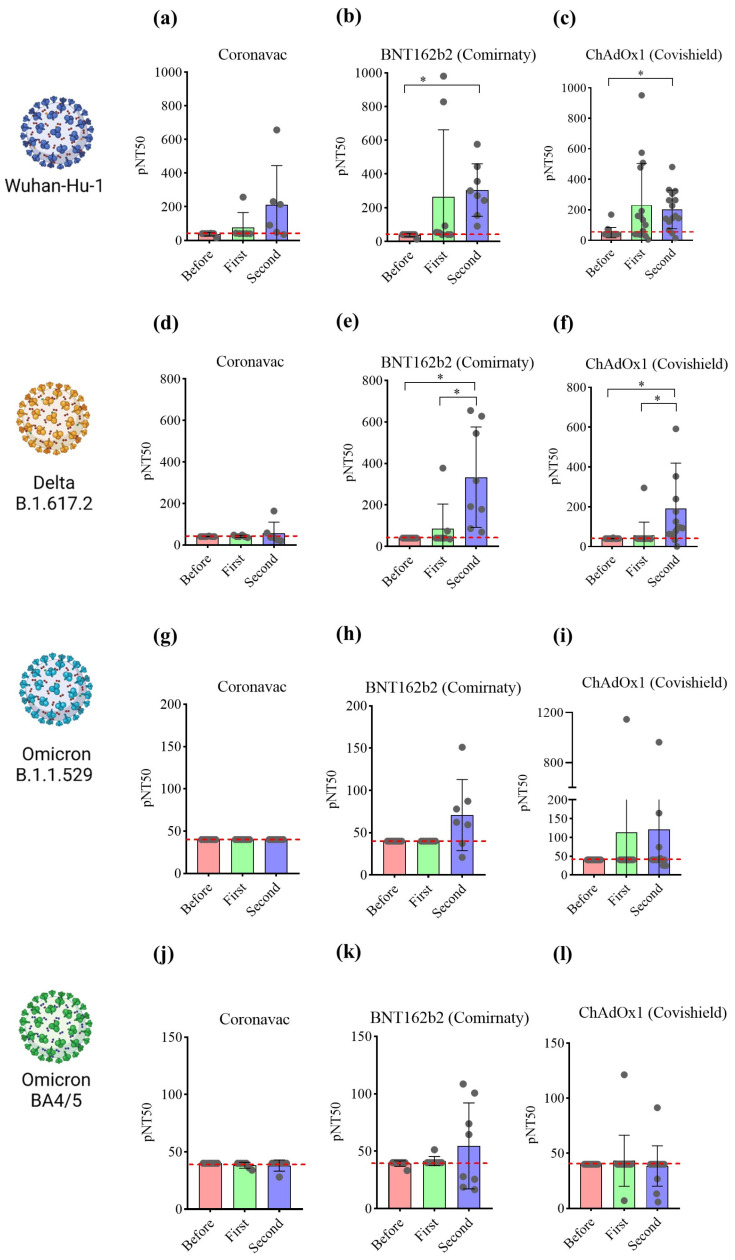
Imaging-based analysis characterizes cohorts based on the vaccine administered and number of doses. Validated samples were diluted and submitted to the neutralization image-based assay. Serum potency (pNT_50_) was determined via nonlinear regression as a function of titers log_10_ and the inhibition of pseudovirus infection. The pseudoviruses produced expressed (**a**–**c**) Wuhan-Hu-1, (**d**–**f**) Delta B.1.617.2, (**g**–**i**) Omicron B.1.1.529, or (**j**–**l**) Omicron B.A4/5 spike proteins. Values considered statistically significant when *p* < 0.05 (*) by using Friedman test, followed by Dunn’s multiple comparisons test, calculated by using GraphPad Prism, version 8.0. Data are shown as mean ± SEM. Circles indicate the volunteers, and red line indicate the lowest pNT_50_ value equal to 40.

**Figure 7 vaccines-12-00236-f007:**
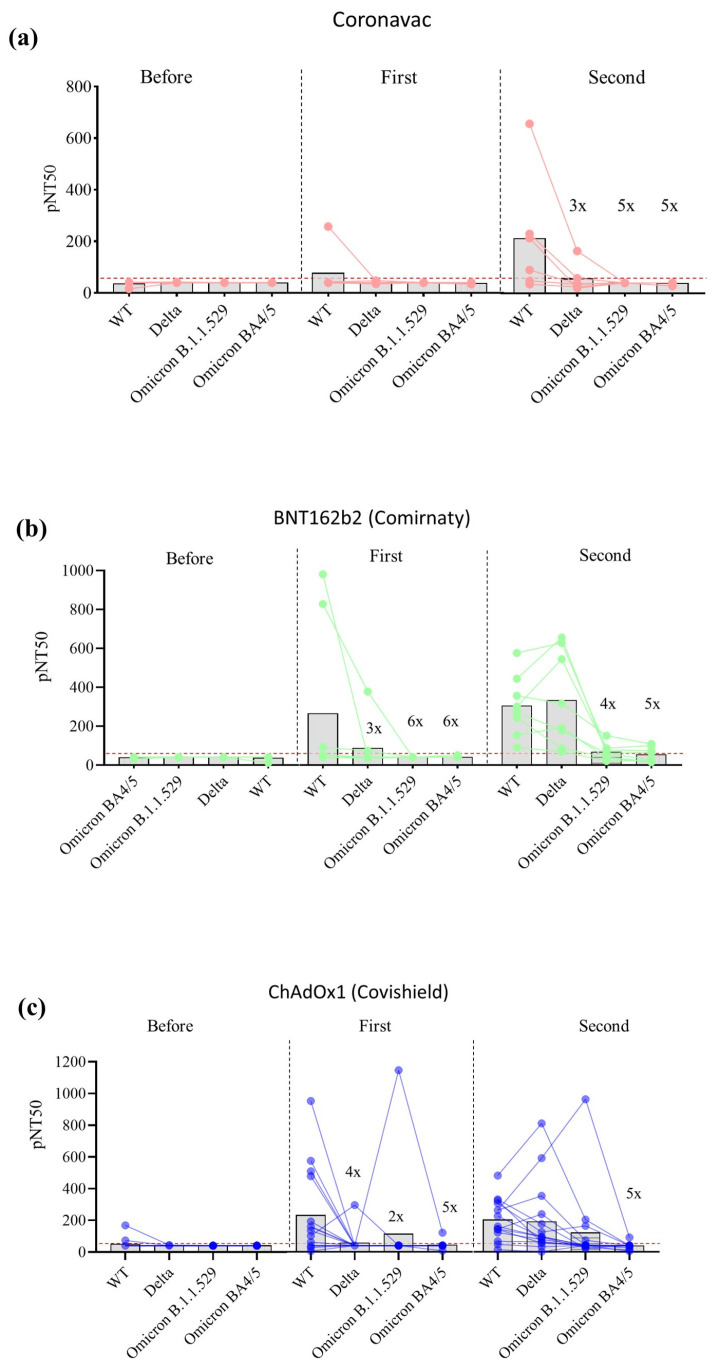
Imaging-based analysis can detect VoC neutralization escape. pNT_50_ values were used to compare serum potency with respect to different VoCs. Graphs show VoC neutralization escape after two doses of (**a**) Coronavac, (**b**) Comirnaty, or (**c**) Covishield. The numbers above the bars (such as 3×, 4×, 5×, and 6×) are the ratio between wild-type and VoCs pNT_50_ and denote the resistance of variants to the serum tested.

## Data Availability

Created data is reported in the manuscript.

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
