# Peer review of "High-Content Imaging-Based Assay for SARS-CoV-2-Neutralizing Antibodies"

_vaccines, 2024, doi:10.3390/vaccines12030236_

Round 1
Reviewer 1 Report
Comments and Suggestions for Authors
The authors have resubmitted their manuscript with some revisions in response to reviewer comments. The manuscript has been improved in several respects and most review comments have been adequately adddressed; however, much of the revisions to the text have been introduced hastily without proofreading.
Consider the paragraph starting at Line 149:
“Repetibily (intra-assay) was determined by one analyst through the performance 149 of three independent experiments (three viral batches) in triplicate. Repetibility 150 (interassay) as determined by two analystes who performed thre independent 151 experiments with five replicates [27]. The coefficient of variation (CV) was calculated 152 by the ration between the standard deviation and the pNT50 mean values. The CV up 153 to 15% was accepted for both precision parameters [28].”
There is no such word as Repetibily; there are words missing; the word three is misspelled, just to mention a few of the errors. This level of writing is unacceptable. It is not the purview of the reviewer to re-write; the authors need to go over all of the revised text with someone who is good at scientific writing. These paragraphs need to be rewritten so that they are comperhensible and grammatical.
Similar problems for the paragraph added at line 244.
Same for paragraph added at line 284.
The term “analyst” is used throughout. What is an analyst? Do they mean analysis?
The authors have added extensive statistical data to the text in the Results section making the text difficult to read. Most of this information can be moved to the figure legends. The text would then be easier to read.
Comments on the Quality of English Languagecomments on quality of English are included in comments in the section above.
Author Response
1- Consider the paragraphs starting at Lines 149, 244 and 284
Authors: Dear reviewer, thank you for these comments. We apologize for our mistakes in the text. It was corrected as you mentioned. You can see the corrected paragraphs starting at line 149, added at line 244 and 284.
2- The term “analyst” is used throughout. What is an analyst? Do they mean analysis?
Authors: We changed this term to researcher(s)
3- The authors have added extensive statistical data to the text in the Results section making the text difficult to read. Most of this information can be moved to the figure legends. The text would then be easier to read.
Authors: We corrected it, as your suggestion. Thank you.
Reviewer 2 Report
Comments and Suggestions for Authors
Manuscript 2889409
Just a few minor errors to be corrected:
Lines 310-1: “Samples from volunteers vaccinated once with CoronaVac presented pNT50 values at the detection limit of 40, as those prior to vaccination.”
Line 355: “(Figures 5B AND 5C).” The second dose of CoronaVac did not significantly increase the neutralization capacity of the sera.
Comments on the Quality of English LanguageLine 148: “Assay precision”
Line 151: “analysts;” “three”
Line 153: “ratio”
Figure S5 is mislabeled.
Author Response
Comments and Suggestions for Authors
- Lines 310-1: “Samples from volunteers vaccinated once with CoronaVac presented pNT50 values at the detection limit of 40, as those prior to vaccination.”
- Line 355: “(Figures 5B AND 5C).” The second dose of CoronaVac did not significantly increase the neutralization capacity of the sera.
- Line 148: “Assay precision”
- Line 151: “analysts;” “three”
- Line 153: “ratio”
- Figure S5 is mislabeled.
Authors: Thank you for the extensive review you did in our work. All the corrections and suggestions contributed to improving the quality of the manuscript. We performed all the corrections you suggested, and you can track it in the text and supplementary file.